# The Neurokinin-1 Receptor: Structure Dynamics and Signaling

**Francisco D. Rodríguez** [1,2,*] and **Rafael Coveñas** [2,3]

1   Department of Biochemistry and Molecular Biology, Faculty of Chemical Sciences, University of Salamanca, 37007 Salamanca, Spain
2   Group GIR USAL: BMD (Bases Moleculares del Desarrollo), University of Salamanca, 37007 Salamanca, Spain
3   Laboratory of Neuroanatomy of the Peptidergic Systems, Institute of Neurosciences of Castilla and León (INCYL), University of Salamanca, 37007 Salamanca, Spain
*   Correspondence: lario@usal.es; Tel.: +34-677510030

**Abstract:** Substance P (SP), the first isolated neuropeptide, belongs to the family of tachykinin peptides and is the natural ligand of neurokinin-1 receptors (NK-1R), also named SP receptors. The undecapeptide activates the receptor after specifically binding to the protein and triggers intracellular signals leading to different biochemical events and subsequent physiological responses. This study reviews the main architectural features of this receptor, its interaction with natural and synthetic ligands, and the functional conformational states adopted after interacting with ligands and effector G proteins. The analysis of the main intracellular signaling pathways turned on by the activation of NK-1 receptors reveals the participation of different proteins supporting metabolic changes and genetic and epigenetic regulations. Furthermore, the analysis of receptor occupancy and receptor downregulation and internalization represents a complex and estimable field for basic and clinical research focused on the role of SP in physiopathology. Profound knowledge of the structural dynamics of NK-1R may help develop and assay new selective synthetic non-peptide antagonists as potential therapeutic agents applied to various pathologies and symptoms.

**Keywords:** substance P; neurokinin-1 receptors (NK-1R); NK-1R antagonists; tachykinins; Gq and Gs-coupled receptors (GPCRs); netupitant; aprepitant; protein structure dynamics

## 1. Introduction

In 1931, Von Euler and Gaddum reported the isolation of a standard preparation in a water-soluble powder (later named substance P) from horse tissues that induced the contraction of isolated rabbit jejunums. It also exerted "a temporary fall of blood pressure in the atropinized rabbit under ether" [1]. Afterward, the powder was characterized as a peptide, isolated from the bovine hypothalamus, and its sequence of amino acids was determined [2,3]. SP is an undecapeptide ($H_2N$-RPKPQQFFGLM-CO-$NH_2$) belonging to the family of tachykinin peptides (a large group of tachykinins and tachykinin-like peptides, including invertebrate tachykinin-related peptides and vertebrate tachykinins) [4]. Tachykinin expression refers to the fast-moving contractions of smooth muscle observed in the presence of SP [5]. Etymologically, the two Greek roots ταχύς and κίνησις (meaning rapid movement) form the word. Tachykinins encompass a family of peptides (more than forty peptides isolated from mammalian and non-mammalian species), including SP, neurokinin A (NKA), and neurokinin B (NKB), which are the three most representative species. The *TAC1* gene on human chromosome 7 encodes four mRNA transcripts translated into four isoforms containing the SP sequence. Additionally, SP may be cleaved into different fragments with biological activity [5].

Depending on the type and localization of neurokinin receptors, tachykinins influence numerous biological and pharmacological effects in all bilaterian organisms, from lower invertebrates to mammals, including humans [6]. Tachykinins have a pleiotropic impact on the central and peripheral nervous system, lungs, liver, skeletal muscle, skin, and

gastrointestinal tract [4]. With other tachykinins, SP plays prominent physiological and pathological roles in pain, inflammation, carcinogenesis and cancer progression, major depression, and hematopoiesis, for example [6].

The actions of tachykinins proceed through the activation of three receptors, namely NK-1R (for which SP shows the highest affinity), NK-2R (for which NKA shows the highest affinity), and NK-3R (for which NKB shows the highest affinity) [7–10]. Neurokinin receptors belong to the family of seven transmembrane G-coupled receptors and signal intracellularly by triggering different mechanisms (described below in this review). The internalization of the ligand–receptor complex and the subsequent enzymatic hydrolysis cease the signal. The three receptors have been pharmacologically characterized, and the affinity, selectivity, and binding properties were assessed using agonists and antagonists in cell membranes and isolated organs [11]. Additionally, their tissue distribution has been evaluated by autoradiography studies (see, for example, their distribution and ontogeny in the central nervous system, CNS [12]). NK-1R, NK-2R, and NK-3R are encoded by the *TACR1, TACR2,* and *TACR3* genes.

The amino acid sequence of a human SP receptor was deduced from the nucleotide sequence of a cDNA. Protein expression in COS-7 cells facilitated the determination of the pharmacological properties of the protein and established that SP showed the highest affinity for the protein consisting of 407 residues and was confirmed as belonging to the G protein-coupled receptor superfamily [13]. Figure 1 presents the primary and secondary structures of the NK-1 receptor; it also shows the N-terminal, C-terminal, as well as the intracellular, extracellular, and transmembrane helical domains drawn in a snake plot.

Early studies on structure–activity relationships established the importance of the specific chemical characteristics of neurokinins that are necessary to trigger their receptors [4]. For example, aromatic or aliphatic residues in the C-terminal region of tachykinins and the amidation of the C-terminal residue to activate the receptor protein were considered essential for receptor activation [14].

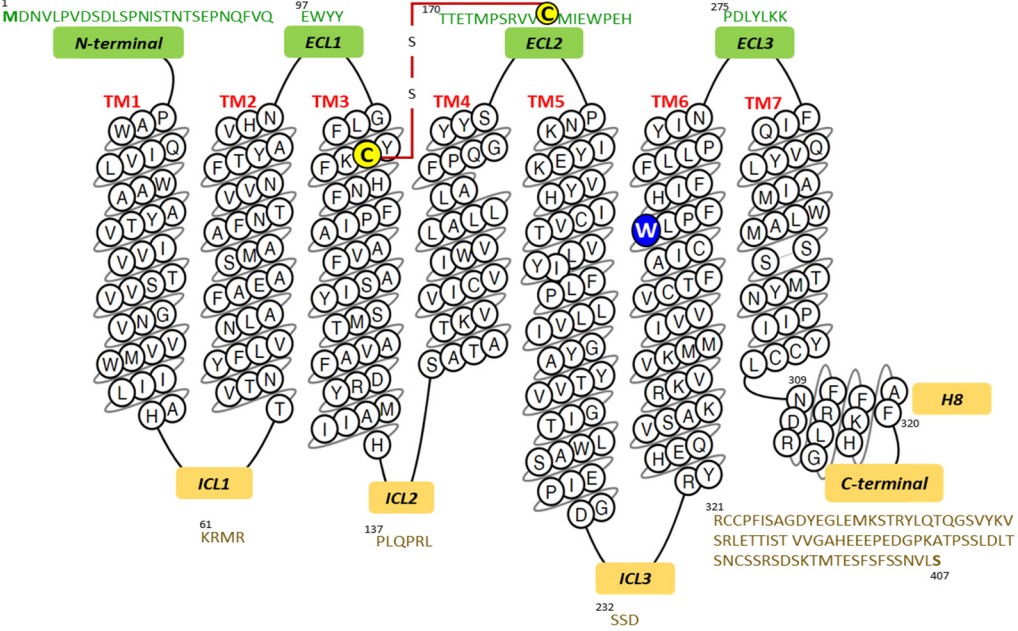

**Figure 1.** Snake plot of the human NK-1 receptor depicting the amino acid sequences of the transmembrane (TM), in the color black; amino-terminal and extracellular loops (ECL), in the color green; intracellular loops, intracellular helix 8 (H8), and carboxy-terminal domains, in the color yellow. The numbers over the amino acid letters indicate their position within the entire receptor sequence from the N-terminal's number 1 residue (Met). The red line depicts a disulfide bridge between Cys180 and

Cys105 (in yellow). The W261$^{6.48}$ residue in blue (superscript numbers correspond to Ballesteros and Weinstein numbering [15]) acts as a "molecular toggle" in the activation of GPCRs (G protein-coupled receptors) [7,8]. Image modified from the GPCRdb database [16,17].

## 2. The Neurokinin-1 Receptor: Structure Dynamics and Interactions with Ligands

The NK-1R belongs to class A of GPCRs (G protein-coupled receptors). This family of seven transmembrane (7TM) receptors shares intrahelical and interhelical hydrogen bonding patterns, helping insert helices into the lipid bilayer and influencing their molecular contacts with other membrane-associated proteins and with agonists and antagonists. The dynamic establishment of hydrogen bonds and atomic displacements within the structure are requirements for their function. The interhelical forces also depend on the membrane's lipid composition [18–20]. Consequently, analyzing atomic events facilitating the displacement of atoms and structures within these membrane proteins is paramount to ascertaining their activity and regulation in physiological and pathological contexts.

The complete form of NK-1R consists of 407 amino acids and the truncated splice variant of 311 amino acids. The truncated isoform has a short cytoplasmic C-terminal domain lacking essential residues responsible for the anchoring of this domain to the internal bilayer of the plasma membrane [21]. Additionally, the C-terminal end of NK-1R full form serves as a substrate for essential modifications such as phosphorylation of specific amino acid residues [22] required for receptor internalization and binding to β-arrestins and other transducers [23,24]. Notably, both isoforms may show differentiated expression, signaling profiles [25], and roles in physiological and pathological processes [26,27]. Given the structural differences between both isoforms, substance P diversifies its activity and extends its influence in agreement with tissue and time-dependent expression of the NK-1R variants [28].

### 2.1. SP Is the Natural Agonist of NK-1 Receptors

SP is the natural ligand of NK-1 receptors together with hemokinin-1 (HK-1) [29–31]. The undecapeptide exhibits flexible random structures in an aqueous environment, and its 3D form is influenced by temperature, pH, and the presence of calcium ions under physiological conditions. The peptide's randomly disordered structure is biologically inactive [32]. Neutron diffraction experiments carried out in bilayers composed of DOPC (dioleylphosphatidylcholine) and DOPG (dioleylphosphatidylglycerol) reported that SP interacts with both zwitterionic and anionic phospholipids. The C-terminal region of SP penetrates the bilayer, whereas the rest sets on the surface of the bilayer [33]. SP residues Pro-4, Gln-5, Phe-7, Phe-8, and Gly-9 determine the selectivity of the binding to the receptor [4,34]. Mutagenesis experiments unveiled important amino acid positions for the accommodation of the peptide to its site within the receptors, namely Asn-85, Asn-89, Tyr-92, and Asn-96, located in TM (transmembrane) domain 2, and Tyr-287 in TM domain 7 [35]. NMR determinations, docking experiments, molecular dynamics simulations, energy minimization, and the use of benzo-Phe-SP [36–38] and the site-specific placement of benzoPhe [39] addressed the possible contacts that the peptide ligand and the receptor establish during the binding action. The proposed models focused on the interaction between SP with the N-terminal domain of the receptor (residues 11 to 21) and with the extracellular loop 2 (ECL2) residues 173-177. Moreover, the C-terminal region of the peptide moved into deeper contact with the transmembrane regions TM5 and TM6 inside the receptor's structure, and the central part of SP adopted a helical conformation that was favorably accommodated within the extracellular loops ECL2 and ECL3 (Figures 1 and 2).

**Figure 2.** Panel (**A**) shows SP's sequence of amino acids and its two-dimensional structure drawn with KingDraw-Free software [40], indicating the N-terminal, central, and C-terminal regions. Within the primary sequence shown, residues in blue are determinants of the selectivity of the ligand for its receptor [4,34]. The positions Phe7 and Phe8 are indicated within the chemical representation of the peptide. The lower panels (**B**,**C**) depict illustrative structures obtained from the Protein Data Bank [41] (corresponding to the conformation of SP in artificial bilayers harboring the NK-1R in the presence of water (panel (**B**)) corresponding to PDB Id. 2KS9 or ganglioside GM1 (panel (**C**)), corresponding to PDB Id. 2KSA) [42]. Both structures were drawn with Mol* free web-based software [43].

In vitro experiments using isotropic bicelles (structures composed of various phospholipids in different proportions) containing the ganglioside GM1 showed that SP assumed a stable turn/helical (extended 310 helix or turn III) conformation that facilitates binding to the membrane receptor. These artificial bicelles were designed and built to mimic the native membrane environment where NK-1 receptors are located in membrane caveolae rich in GM1 [42]. The undecapeptide shows a flexible and disordered conformation in the presence of water (Figure 2B; PDB Id. 2KS9). However, SP adopted a defined and ordered structure, corresponding to the bioactive conformation that interacts with NK-1 receptors in the presence of ganglioside GM1 (Figure 2C; PDB Id. 2KSA). Additionally, the assessment of hydrogen bonding with VADAR (Volume Area Dihedral Angle Reporter) [44] and the determination of chemical shifts in bicelles by $^1$H-NMR in water alone or the presence of GM1 indicated that the C-terminal region of the peptide adopts a more defined conformation. In contrast, the N-terminal domain has a more flexible structure [42] (Figure 2B, PDB Id 2KS9 and C PDB Id. 2KSA).

Cryo-electron microscopy studies revealed an active state of the NK-1 receptor complexed with Gq and Gs heterotrimeric proteins and bound to SP [45]. The occurrence of cholesterol molecules in the active form of the receptor (Figure 3, left panel) reinforces the reported importance of the organization of the plasma membrane with lipid rafts and caveolae where NK-1 receptors sit and recruit intracellular effectors after binding to SP [46].

The binding of SP to the receptor complexed with a heterotrimeric G protein indicates that the C-terminal region of the undecapeptide binds the receptor protein by interacting with its transmembrane core. In contrast, the N-terminal part extends towards the extracellular region (Figure 3, right panel) [45]. Additionally, the cryo-EM structural determination of the NK-1R-SP-G $_{s/q70}$ complex showed that the N-terminal segment of SP establishes close contact with the extracellular loop 2 (ECL2) and the N-terminal portions of the receptor; the C-terminal region of SP comes into contacts with receptor residues N85 (in helix 2), N89 (in helix 2), H108 (in helix 3) and Y287 (in helix 7) through a network of hydrogen bonds [47]. The binding site of SP is different from the orthosteric pocket that NK-1R antagonists occupy, where the region of the receptor overlapped by the occupancy of SP and antagonists is scarce. For example, netupitant only shares a binding location with the amidated methionine at position 11 of the undecapeptide [47].

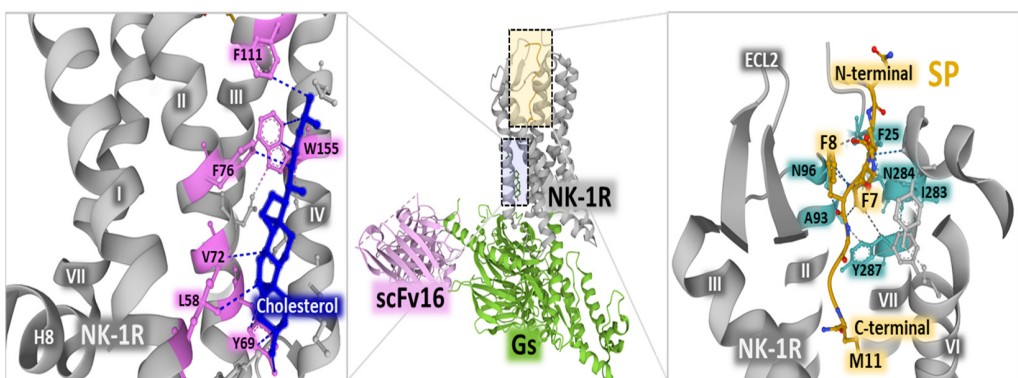

**Figure 3.** The central panel represents NK-1R (gray) complexed with Gs heterotrimeric protein (green) and scFv16 (single-chain variable antibody fragment used to stabilize the complex) (light pink). The panel on the left shows the position of cholesterol (dark blue) within the receptor helices, indicating weak interactions with receptor residues (pink) with dashed lines. The right panel depicts SP's position (yellow) within the receptor core. The C-terminal, N-terminal, and SP amino acids F7, F8, and M11 are indicated. Dashed lines represent weak interactions of the undecapeptide with NK-1R residues (light blue). The figure also shows the positioning of NK-1R, helices, and extracellular loop 2 (ECL2). The structures are from the Protein Data Bank [41] (PDB, corresponding to PDB Id. 7P02) [45], drawn with Mol* free web-based software [43].

Examining the molecular structure of NK receptor subtypes and the interaction with their ligands brings attention to their unique role in physiological and pathological activities. The mode of interaction of agonists and antagonists with their respective receptors explains their selectivity, efficacy, and potency. A recent study [48] compared the structures of SP bound to the complex NK-1R-Gq and neurokinin A (NKA) attached to NK-1R-Gq, at the atomic level. Interestingly, apparent differences concerning the interaction of the N-termini of neurokinins with the extracellular loop 2 (ECL) and other domains of their respective receptor explain different binding kinetics and physiological responses. This knowledge reports valuable information to understand structure–activity relationships and to design new compounds capable of modulating the activity of NK receptors in a specific and effective manner.

### 2.2. Antagonists of NK-1 Receptors

Early studies of crystal structures of NK-1R antagonists analyzed possible interactions of different pharmacophores (for example, quinuclidine derivatives, L-tryptophan benzyl esters, or piperidine rings) with the environment to obtain helpful information concerning their real contacts and binding capabilities to the receptor residues [49]. The analysis pointed to the importance of probable intermolecular interactions with some residues, including N165, H265, F268, and H197. However, determining the complex geometries between ligands and proteins requires the receptor protein's presence. Later crystallographic

and NMR structural studies provided more accurate information concerning ligand interactions with the receptor protein.

The synthesis of neurokinin receptor antagonists provided advances for studying the three types of NK receptors concerning binding, selectivity, and blocking activity. These compounds are also valuable for determining the structural properties and dynamics of the receptors by attaining crystal structures of the complex receptor-antagonist that permit unraveling the interactions with amino acid residues and defining three-dimensional orthosteric pockets with precision. The co-crystallization of the human NK-1 receptor fused with the thermostable PGS (*Pyrococcus abysii* glycogen synthase) with bound antagonists CP-99,994, aprepitant, and netupitant generated diffracting crystals that showed the structure of a type A GPCR (G protein-coupled receptor) in an inactive conformation with seven transmembrane domains, an extracellular N-terminal domain, three extracellular loops, and three intracellular loops, as well as an intracellular helix (helix 8) parallel to the plasma membrane, and a C-terminal intracellular domain [16,17]. A conserved disulfide bridge between Cys180 in extracellular loop two and Cys105 in transmembrane (TM) domain 3 connects the extracellular domain 2 (ECL2) with the transmembrane domain 3 (TM3) (Figure 1). The work of Schöppe et al. offers detailed information regarding the interatomic relationships between the receptor and the antagonist compounds. For example, netupitant sits in a hydrophobic cleft, establishing some essential weak interaction with residues within the receptor (Figure 4A,D). A longer linker in netupitant (compared with similar antagonists) between the di-trifluoromethylphenyl structure and the methylphenyl group approximates the compound towards F268 and H197, thus promoting π–π interactions between the methylphenyl ring and the imidazole of histidine (Figure 4D). These aromatic positions significantly influence the binding affinity of the antagonist netupitant for the NK-1 receptor. The accommodation of netupitant in a hydrophobic cavity within the NK-1 receptors creates unique weak interactions and anchors helices V and VI, leading to an inactivated protein conformation that impairs its function (see Figure 4D). In crystallographic structures of mutated E78D NK-1R bound to aprepitant and wild-type NK-1R attached to L-760735, a network of weak interactions with crucial residues, including H197, E193, F264, or W261, secures aprepitant (Figure 4B,E) [50] and L-760735 (Figure 4C,F) [51] into a hydrophobic cleft similar to the cavity occupied by netupitant where the antagonist orients perpendicularly to the plasma membrane, establishing interactions with TM3, TM5, and TM6 [51]. A recent study with carbohydrate (as a central scaffold)-based NK-1R antagonists revealed that the antagonist pocket of the receptor stabilizes the structure by arranging hydrophilic interactions between the pyranose moiety and amino acids Asn 109 and His108, thus improving and securing the stability of the binding and the affinity and selectivity of the compounds [52].

The structural analysis provides the basis for understanding the role of some residue positions that bear crucial functions. For example, Figure 4D–F and Figure 1 show the place of residue W261, situated in TM6, lying in the inferior region of the antagonist pocket [51]. The displacement of this conserved tryptophan functions as a rotameric switch in the activation of class A GPCRs [8] and may act as an essential component of an allosteric interface necessary for regulating receptor function [7]. The indole ring of W261 interacts with the trifluoromethyl-benzyl group of the antagonists and stabilizes the complex (Figure 4F).

*2.3. Insurmountable Antagonism of NK-1 Receptors*

The displacement of agonist dose–response curves to the right and the reduction of the maximal agonist response define insurmountable antagonism. This antagonism correlates with structural adjustments between the antagonist drug and the receptor protein. Many different mechanisms may account for this antagonist effect, for instance, the stability and binding kinetics of the antagonist–receptor complex. Additionally, slow change rates between receptor conformations, allosteric regulation, or the internalization of the protein in the antagonist–receptor complex may explain this phenomenon. Recent co-crystallization

and X-ray diffraction studies with the selective antagonists aprepitant, netupitant, and NK-1R show that when the antagonists bind to the NK-1 R, a new hydrogen bonding network ties up the transmembrane helices TMVI and TMVII to become closer.

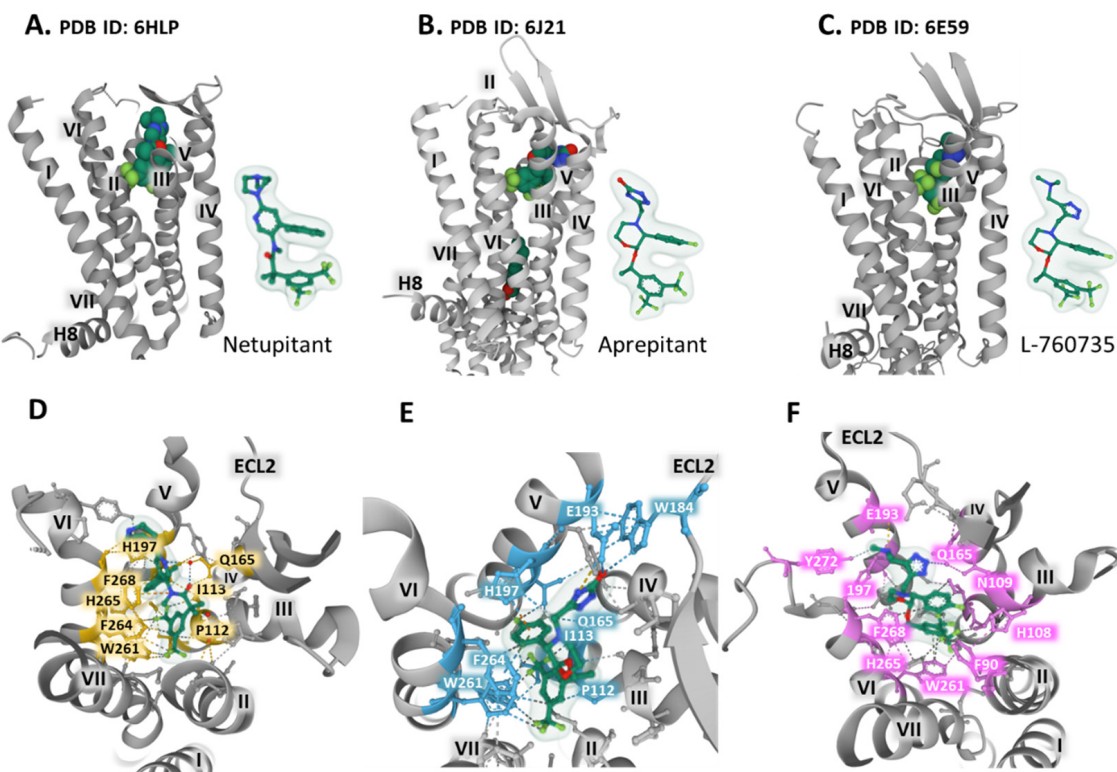

**Figure 4.** The antagonists (space fill representation) nenupitant (**A,D**), aprepitant (**B,E**), and L-760735 (**C,F**) occupy an NK-1 receptor hydrophobic pocket. The upper pictures, (**A–C**), present a view of the co-crystalized complex parallel to the plasma membrane plane, together with a representation of the antagonists' ball and stick and the Gaussian volume (light green). The lower panels (**D–F**) depict a more detailed view of the complex observed from the extracellular space and relate to the (**A–C**) structures. Dashed lines represent weak antagonist bonding with receptor residues (yellow for netupitant, blue for aprepitant, and pink for L-760735). Transmembrane (TM) helices appear in roman numbers. ECL2 is extracellular loop 2. The figure displays the structures obtained from the Protein Data Bank [41], corresponding to NK-1R bound to netupitant (PDB Id. 2HLP, A and D), the E78$^{2.50}$D mutant bound to aprepitant (PDB Id. 6J21, B and E) [50] and NK-1R attached to L-760735 (PDB Id. 6E59, C and F) [51], viewed with the Mol* free web-based software [43].

Consequently, the whole structure loses conformational flexibility. This structural restriction imposed by the interaction of the antagonist with NK-1R may explain the observed insurmountable antagonism exerted by both aprepitant and netupitant in experimental settings. A comparison of crystal structures of NK-1R bound to insurmountable antagonists and structures of NK-1R bound to G$\alpha$ and Gq proteins [45] revealed that the receptor's conformation with the antagonists prevents SP binding due to steric hindrance contacting ECL2 and the imposed difficulty for the movement of TMVI to facilitate receptor activation.

### 2.4. The Amino Acid Position 2.50 in NK-1 Receptors

In contrast to the majority of GPCRs, where the 2.50 position (according to Ballesteros and Weinstein numbering, [15]) is aspartic acid (D78$^{2.50}$), the NK-1R has a glutamic acid residue (E78$^{2.50}$). The variant relates to the absence of constitutive NK-1R signaling [39]. The NK-1 receptor bound to netupitant reveals a network of direct and water-mediated hydrogen bonds established by E78$^{2.50}$, mostly with residues S119$^{3.39}$, N301$^{7.49}$, and N50$^{1.50}$. (Figure 5). These weak bonds are responsible for the inactive conformation of the receptor.

The extended carbon atom chain in glutamic acid situates the residue in an advantageous position compared with aspartic acid, thus permitting the creation of interactions similar to those observed between $D78^{2.50}$ and sodium in other GPCRs, where partially hydrated sodium ions act as negative allosteric modulators (Figure 6) [53,54]. Mutational studies provided evidence of a unique interface formed by residues $E78^{2.50}$, $S119^{3.39}$, and $N30^{7.49}$ that triggers differential signaling pathways [39,55]. Moreover, in a recent report, mutations at the wild-type amino acid position $E78^{2.50}$ to aspartic acid ($D78^{2.50}$) or asparagine ($N78^{2.50}$) revealed that the hydrogen bond between $E/D78^{2.50}$ and $N301^{7.49}$ disappears when asparagine occupies the $X^{2.50}$ position (Figure 5B,C). For that reason, a consequent displacement of a short sequence where residue $N301^{7.49}$ sits may affect essential interactions for receptor activation [50]. Additionally, the hydrogen bond between the $X^{2.50}$ position and $S119^{3.39}$ is less robust with either aspartic acid or asparagine at position 2.50 than glutamic acid (Figure 5). The studies mentioned above indicate that the $X^{2.50}$, $X^{3.39}$, and $X^{7.49}$ posts form an interface that plays an essential role in the activation states of the receptors dependent on weak interactions with neighboring residues [39].

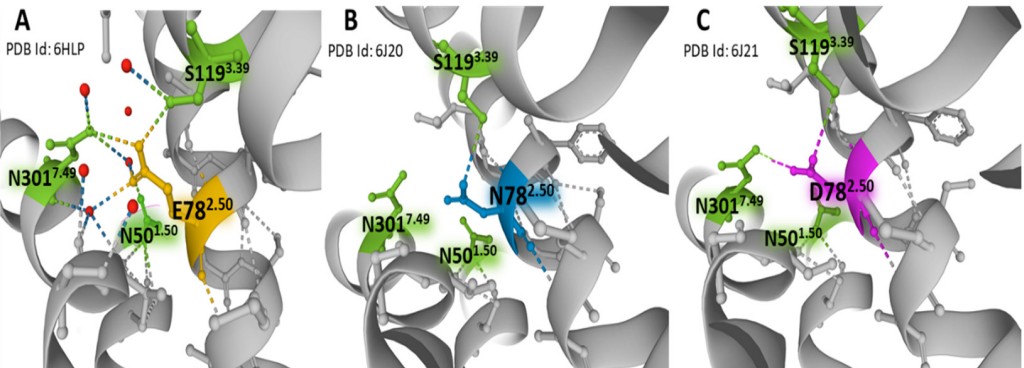

**Figure 5.** Weak bonding interactions of different residues occupying the $X^{2.50}$ position with neighboring $S119^{3.39}$, $N301^{7.49}$, and $N50^{1.50}$ in the structure of NK-1R. Panel (**A**) represents the position of $E78^{2.50}$ in a wild-type NK-1R bound to antagonist netupitant (PDB Id: 6HLP), defining the basal inactive state of NK-1 receptors. Panel (**B**) represents the position of $N78^{2.50}$ in a mutated NK-1R bound to antagonist aprepitant (PDB Id: 6J20) [50]. Panel (**C**) illustrates the situation of $D78^{2.50}$ in a mutated NK-1R attached to the antagonist aprepitant (PDB Id: 6J21) [50]. Receptor helices are depicted in gray. The figures are from the Protein Data Bank [41] and were drawn with the Mol* free web-based software [43].

The work mentioned above points to a specific structure of the ligand SP and a defined membrane environment to successfully bind to the receptor. Recent structural studies using specific NK-1R antagonists precisely show the properties of the receptors, their binding to ligands and G proteins, and the dynamic movements that secure their effective and fine-tuned function. Defining specific sites for antagonists facilitates the design of new structures that may improve the receptors' binding effectivity and blocking capacity. The evidence provided may help design antagonists more efficaciously and in a way that is specifically suitable for application in numerous pathologies, from pain relief to the treatment of alcohol use disorders, cancer, emesis, and depression [6]. Designed antagonists attached to hydrophobic structures or new compounds without this requirement, targeting NK-1 endosomal signaling, may represent another sound approach to controlling NK-1R function. With this goal in mind, a consideration of the conformational fitting induced by antagonists on the receptor structure is paramount, and research in this field may report on new effective drugs for the treatment of different pathologies or the alleviation of symptoms, including pain [24].

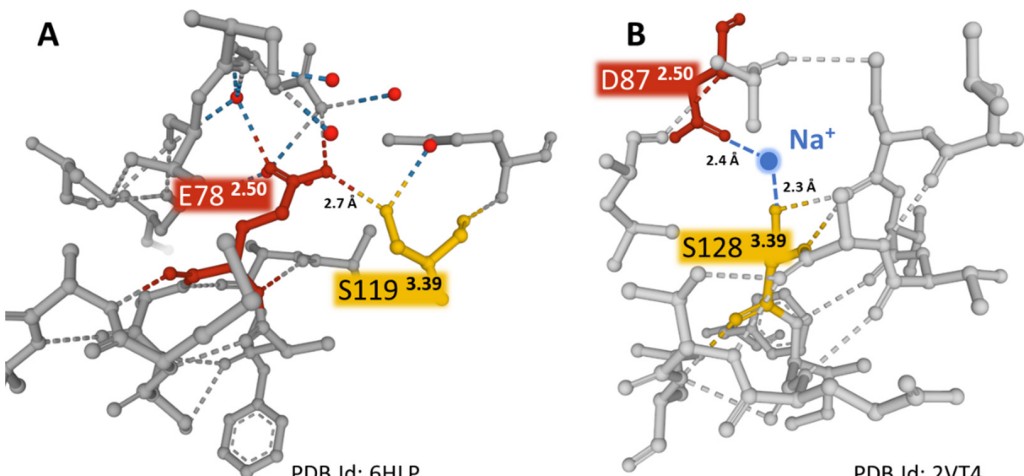

**Figure 6.** The weak interactions (dashed lines) between positions 2.50 and 3.39 in two different GPCRs. Panel (**A**) shows E78[2.50] directly interacting with S119[3.39] in the NK-1R (PDB Id 6HLP). Panel (**B**) depicts the interaction of D87[2.50] with S128[3.39] coordinated with a sodium ion in the turkey β-1 adrenergic receptor (PDB Id 2VT4) [56]. The sodium ion is blue, and the water molecules are red-colored spheres. The figures were obtained from the Protein Data Bank [41] and drawn with the Mol* free web-based software [43].

## 3. The Neurokinin-1 Receptor: Structure Dynamics and Signaling

### 3.1. G Protein-Coupled Receptors (GPCRs)

G protein-coupled receptors (GPCRs) are abundant and ubiquitous membrane proteins that respond to stimuli from photons to ions, small organic compounds, peptides, and macromolecules. They are responsible for multiple cellular responses and share a serpentine architecture with a package of seven transmembrane helices, alternate intracellular and extracellular loops, the C-terminus lying in the cytoplasm, and the N-terminus expanding towards the extracellular space [17,57–62]. GPCRs activate different G protein and β-arrestin-dependent mechanisms that control many physiological and pathological outcomes [63] related to immune responses, mood, cognition, blood pressure control, pain control, taste, vision, and olfaction [61,64–66]. Signaling can also occur via endosomal membranes that facilitate the activation of localized signals [67–69]. The group is subdivided into classes considering their sequences' similarities and structural features: A (rhodopsin), B (secretin), B2 (adhesion), C (glutamate receptors), D1 (Ste2-like fungal pheromone), family F (frizzled, FZD, and smoothened, SMO), and T (taste) [16,65]. Extensive and meticulous structural studies of GPCRs provide essential information that allows for a better understanding of their function. They also improved drug design based on the knowledge of protein conformational movements linked to activation and inactivation states and biased agonist signaling [57,65,70–72]. X-ray crystallography studies, molecular dynamics simulations, and the mutational analysis of allosteric interfaces with hydrogen bonding networks in transmembrane helices determine states of activity that trigger intracellular signaling [61,62,73–75]. Updated information on the structural studies of GPCRs can be found in the GPCRs database [16,17]. GPCRs serve as a target for pharmacological intervention. Approximately 35% of officially approved drugs target GPCRs directly or GPCR-associated proteins [76,77]. The list may grow more prominent due to new methods of drug design exploration based on novel receptor regulation concepts, allosteric changes, biased signaling, and receptor heteromerization [78–81]. Additionally, post-translational modifications (phosphorylation, glycosylation, ubiquitination, and palmitoylation) influence receptor folding, dimerization, and function [82]. Soundly, these modifications should also be focused on when studying the pharmacological manipulation of GPCRs.

Class A GPCRs (to which NK-1 receptors belong) are the most abundant and studied. They have an enormous influence on human physiology and pathology. Agonist binding to these receptors activates G protein recruitment by moving helix number 6 towards

the exterior of the receptor bundle caused by a common activation mechanism affecting specific residues [72]. The first structures defined with X-ray diffraction data were bovine rhodopsin [83] and human β-adrenergic receptors [67,84]. More than a hundred structures of GPCRs have been resolved to date. Conserved sequence motifs, such as $D(E)R^{3.50}Y$ in TM3 and $D/E^{6.30}$ in TM6, are responsible for the inactivation state of the receptors by establishing an ionic bridge tightening TM3 and TM6 together, thus preventing the flexible outward movement of TM6 necessary for activation and signaling.

### 3.2. NK-1R Signaling Pathways

The NK-1 receptor belongs to class A of GPCRs. Upon activation, it signals through different pathways depending on its intrinsic performance, topology within the membrane bilayer, tissue localization, and the type of ligand. Agonist ligands of NK-1R include SP, SP analogs, and neurokinin A (NKA) [85–88]. The main signaling routes activate Gq and Gs heterotrimeric proteins, causing intracellular variations in second messenger concentrations, such as calcium, cAMP, or IP3 [89,90]. Figure 7 depicts the main signaling pathways activated by NK-1 receptors, leading to operative protein phosphorylation cascades and metabolic regulations responsible for physiological and pathological consequences.

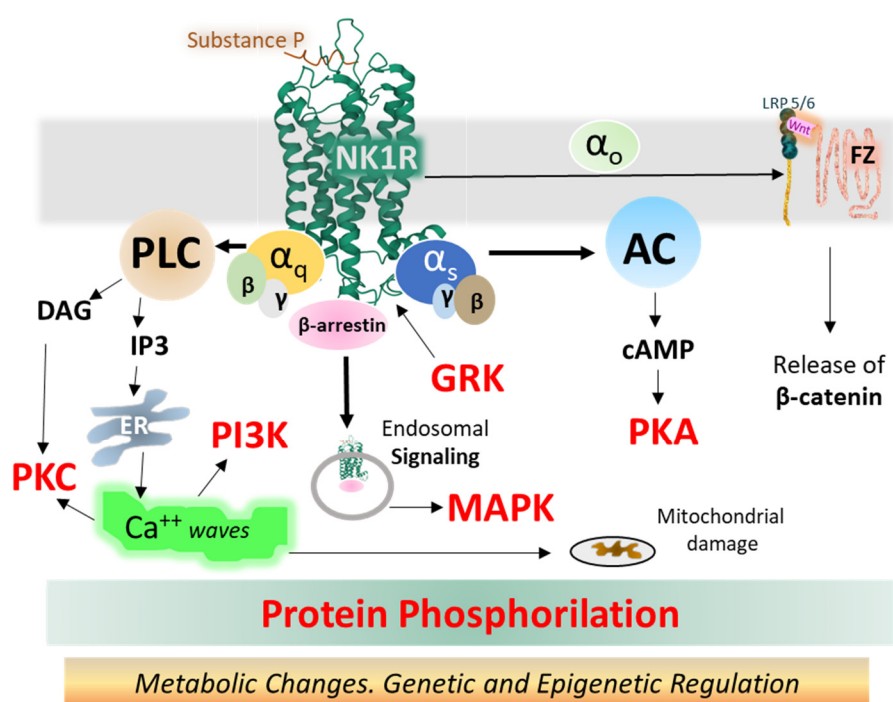

**Figure 7.** Intracellular NK-1R signaling pathways turned on after binding SP and activating different alpha subunits (Gq and Gs, mainly) and beta-gamma dimers of heterotrimeric G proteins (the structure of NK-1R in green is from the Protein Data Bank [30], PDB Id. 2KS9) [31]. Abbreviations: AC (adenylyl cyclase), cAMP (cyclic 3′-5′ adenosine monophosphate), DAG (diacylglycerol), ER (endoplasmic reticulum), FZ (frizzled), GRK (G protein-coupled receptor kinase), IP3 (inositol 1,4,5 trisphosphate), LRP 5/6 (low-density lipoprotein receptor-related protein), MAPK (mitogen-activated protein kinase), NK-1R (neurokinin-1 receptor), PI3K (phosphatidylinositol 3-kinase), PKA (protein kinase A), PKC (protein kinase C), PLC (phospholipase C), and Wnt (wingless-related integration site) [26,89,91–99].

The observed differential activation of NK-1R signaling cascades by agonists may be caused by structural coupling between the receptor and agonist, where efficacy and selec-

tivity lie in different architectural settings. The interface formed by amino acid positions $E^{2.50}$, $S^{3.39}$, and $N^{7.49}$ sets in a net of hydrogen bonds (Figure 5) and facilitates receptor conformations leading to biased receptor activation (Gs, Gq, and β-arrestin) [55]. In this hydrogen bond network built by the amino acid lateral chains, the alanine substitution of $E^{2.50}$ abolished SP-activated Gq signaling measured by the accumulation of inositol 1, 4,5 trisphosphate ($IP_3$). However, the alanine substitution of $N^{7.49}$ and $S^{3.39}$ had a marginal effect on Gq stimulation.

SP and neurokinin A (NKA) and neurokinin B (NKB) bind NK-1 receptors; however, differences in the charge distribution and amino acid composition of the peptides and their interaction with the anionic surface formed by acidic residues in extracellular loops 2 and 3 favor SP binding (higher affinity) over the other two tachykinins [45]. The structural resolution of NK-1 receptors bound to Gs and Gq proteins contributed to determining that the receptor similarly binds to both G proteins [45]. However, the binding to Gq is slightly favored and accords with the observed preference of Gq over Gs signaling [100]. Mutagenesis experiments and the structural dynamics analysis of ligand–receptor interactions [47,85,86] showed that SP and its truncated form SP6-11 (lacking the N-terminal region of the undecapeptide) triggered calcium signaling with similar potency when bound to wild-type NK-1 receptors. However, NKA and SP6-11 induce a weaker cAMP accumulation response than SP. Additionally, experiments with NK-1R bearing conservative mutations of amino acids establishing hydrogen bonding nets deeper inside the pocket ($N85^{2.57}$, $N89^{2.61}$, $H108^{3.28}$, and $Y287^{7.35}$) with the C-terminal region of SP (Figure 8) did not affect the potency and maximal response of the undecapeptide. In contrast, truncated SP6-11-dependent responses significantly decreased [47].

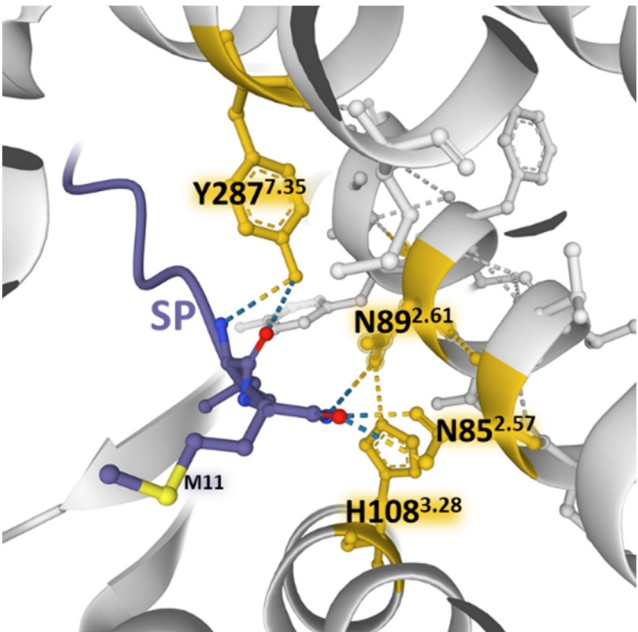

**Figure 8.** Weak bonding interactions of different amino acid residues (yellow) coming into contact with the C-terminal region of SP (blue) deep inside the orthosteric pocket of the NK-1 receptor (gray) (PDB Id: 7RMG [47]). Conservative mutations of the depicted positions do not affect the SP-induced increase in intracellular $Ca^{2+}$ concentrations. The figure also indicates residue M11 of the undecapeptide. The structure is from the Protein Data Bank [41], drawn with the Mol* software [43].

The findings point out that the calcium signaling depended on the interaction of the C-terminal region of SP with the receptor. However, the findings also showed that contact between the central and N-terminal parts of SP with the extracellular domains of NK-1R (ECL2 and N-terminus area) is essential for signaling through Gs and not only for ligand recognition and selectivity. Molecular dynamics simulations reported that the

residues F7 and M11 in SP6-11-truncated peptide adopt more orientations than F7 and M11 in SP undecapeptide and establish differential contacts with TM7 and TM6 of the NK-1R [47]. It is worth noting that preferential Gq activation obtained with SP6-11 reinforces the assumption that connections between the N-terminal region of SP with the exterior of NK-1R determined receptor activation-dependent signaling.

Other GPCRs also exhibit biased agonism responsible for activating singular intracellular metabolic reactions, producing a signaling signature at a given time and in a particular target cell [74,101].

The truncated NK-1 receptor isoform lacks the intracellular C-terminal region [21,102]. Full (NK-1R-F) and truncated NK-1 (NK-1R-T) receptor isoforms display functional differences affecting intracellular signaling [26]. The exclusion of the C-terminal tail impairs receptor internalization, making the protein resistant to desensitization [21]. Additionally, the interaction with transducer G proteins is weaker, and calcium release, protein kinase C (PKC) activation, and other phosphorylation events appear delayed [5,25,103].

The so-called canonical Wnt/β-catenin signaling pathways encompass a group of membrane-bound and intracellular proteins, and the route is closely related to the control of immunomodulation and cell proliferation. The activation of the FZ/LRP 5/6 complex (frizzled/low-density lipoprotein receptor-related protein 5/6) by secreted cysteine-rich glycoproteins Wnt (wingless-related integration site) (see Figure 7) mobilizes an effector G protein, disrupts the Axin-GSK3-APC-CK1 complex and impedes the phosphorylation of β-catenin by the kinases GSK-3β and CK1 (casein kinase 1). Consequently, β-catenin accumulates in the cytosol and moves to the nucleus, where it associates with TCFs/LEF (T-cell factor/lymphoid-enhancer-binding factor) proteins and other transcription factors and co-triggers the transcription of target genes by binding to specific promoter sequences [98,104–107]. β-catenin may also translocate to the membrane and stabilize it by building cell adhesion complexes [107,108].

Experimental studies using agonists and antagonists of NK-1R showed that the agonist turning-on of NK-1R induced an increased expression of β-catenin and GSK3β and diminished the expression of DKK1 (Dickkopf 1), an inhibitor of Wnt [109]. Additionally, selective antagonists of NK-1 receptors inhibited cell proliferation in several experimental scenes by influencing Wnt/β-catenin pathways [91,108,110,111].

The NK-1R system controls numerous intracellular processes, including cell growth, cytoskeletal organization, metabolic reaction, or gene expression through biochemical reactions that contribute to cell homeostasis [112]. Malfunction of the receptor induces dysregulation of intracellular signaling cascades, and disease may appear. NK-1R-triggered signaling complexes participating in cell disruption led to inflammation, calcium overloads, altered cell metabolism and growth, and disrupted cell migration conveying to cancer, pain, psychiatric disorders, and other pathologies. Molecular complexes responsible include the Wnt/β-catenin [91], β-arrestins scaffolds [112], JNK and p38/MAPK [113], NF-kB [113,114], PKCδ/ERK/P65 [115], or metalloproteases MMP-2/MMP-9 [116], to mention a few (Figure 7).

Aside from fine structural studies revealing activated and inactivated conformations of the receptors induced by movements of receptor domains, determining the binding kinetics of both agonist and antagonist ligands deserves attention to ascertain signal transduction outcomes and determine new drugs' specificity, efficacy, modulation, and potency [117].

## 4. Conclusions and Perspectives

State agencies have only approved using human NK-1R antagonists for the treatment of chemotherapy-induced nausea and vomiting [118]. Basic experimental research provided much information on the potential efficacy of NK-1R antagonists in other pathological entities related to substance abuse, pain, some types of cancer, and psychiatric disorders, among others [119–123]. The drawbacks have come from the poor efficacy observed in some clinical assays, possibly associated with low receptor occupancy or other factors, and the need to simultaneously target different types of NK receptors for specific pathologies

to attain effectiveness [93]. Receptor occupancy determines drug responses [124]. It is an important issue when considering using NK-1R antagonists in therapy, from psychiatric disorders to malignancies [125,126]. Clinical trials that used NK-1R antagonists to treat individuals with major depressive disorder (MDD) [127] reported low efficacy results, leading to a discontinuation of the studies. A possible explanation for the negative outcomes appeared associated with the observation that only receptor occupancy close to 100% determined the efficacy of NK-1R antagonists [122,128]. Consequently, contemplating dose, route of administration, pharmacokinetics, and tissue penetration to attain required receptor occupancy, the design of specific antagonists that may interfere with activated states of the receptors, and careful selection of patients, demands additional basic and clinical research [125,129].

The accurate determination of structural changes defined by agonist and antagonist contacts at the atomic level governing selectivity, potency, and efficacy is essential for detecting NK-1 receptors to block or lessen given signaling routes and alleviate the effects of NK-1R-dependent malfunction. Biased agonist activation through the same receptor structure that triggers different signals opens new lines for the design of drugs in a way that is more specific, more effective, and addressed to tackling certain disorders related to particular signaling pathways [47,55,101]. Additionally, the heteromerization of NK-1 receptors is a mechanism worth exploring to design specific drugs that modulate the action of the receptor complexes [81]. Furthermore, the role of the NK-1R truncated isoform merits attention in the design of new drugs.

Refined structural determinations and basal functional research concerning the interaction of NK-1 receptors with proteins involved in cellular signaling and cross-talk pathways are necessary to unveil their participation in numerous processes such as cell proliferation, inflammation, melanogenesis, pain control, and mood and stress-related disorders and to design modulation strategies that would eventually permit targeted therapies.

**Author Contributions:** Conceptualization, design, bibliography analysis, draft preparation, writing and review, F.D.R. and R.C. All authors have read and agreed to the published version of the manuscript.

**Funding:** This research received no external funding.

**Acknowledgments:** Support from Programa XIII de la Universidad de Salamanca to the GIR group (Bases Moleculares del Desarrollo) is acknowledged.

**Conflicts of Interest:** The authors declare no conflict of interest.

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
