# Peer review of "The Neurokinin-1 Receptor: Structure Dynamics and Signaling"

_2813-2564, doi:10.3390/receptors1010004_

Round 1

Reviewer 1 Report

The submitted manuscript by Rodriguez and Covenas is a comprehensive and well-written review of the receptor structure and signaling mechanisms of the neurokinin-1 receptor.  The authors cover the known structural motifs and protein level interactions that influence the receptor function and agonist binding.  They then cover intracellular signaling mechanisms that are activated by agonism of this receptor.  Overall, the review is detailed, well organized, and of interest.  I have only minor comments for revision.  Please see specific items below.

-Little space in the review is devoted to the two major isoforms of the receptor (full length and truncated).  A more thorough coverage of this aspect would strengthen the manuscript.

-Figure 7 is rather busy.  I would suggest focusing on the two most commonly activated intracellular mechanisms of the NK1R (Gq and Gs pathways).

-The Conclusions section mentions receptor occupancy and its relation to clinical efficacy.  A more in depth discussion of this would be valuable.  What is known about high occupancy NK1R antagonists on a molecular level?  What makes these particular drugs effective interactors with NK1R binding sites? 

Author Response

Please find enclosed our responses in the attached file.

Reviewer 2 Report

  • This review presents a comprenhensive summary of current and past literature regarding the neurokinin receptor 1 structure and signalling. The topic is particularly relevant to the GPCR field because recent findings from NK1R inactive and active structures have not been discussed elsewhere. The authors covered the topic well and recognised structural dynamic features that could be of relevance for signalling. 
  • The topic has not been covered or summarised in a recent review.
  • The text and figures are clear and the language used is approapriate and easy to understand.
  • The references used are relevant and recent and does not include an excess of self-citations. 
    Although this review focuses on the structural features of the NK1R, the authors might consider the inclusion of a reference to the recent NKA-NK2R-Gq structure (Sun et al. Cell Discovery (2022) 8:72). Particularly because the differences in ECL2 and interaction with the N-terminal residues of tachykinin might play an important role in receptor subtype selectivity. 
  • Perhaps, section 3.2 NK1R signalling pathways could be strengthen if it included references that linked specific signaling pathways to physiological/pathological conditions. 
  • The statements and conclusions drawn are coherent and supported by the listed citations.
  • The figures and images are appropriate. 
    Figure 7 is a little bit busy with very small text. Please, consider editing and removing elements that are not absolutely necessary for the interpretation of the information presented.

Author Response

Please find enclosed our responses in the attached file
